# OpenReview forum: "Feedback Forensics: A Toolkit to Measure AI Personality"
_ICLR.cc/2026/Conference — Submitted to ICLR 2026_

### Official Review · Reviewer_KruL · 2025-10-28

**Soundness:** 3
**Presentation:** 2
**Contribution:** 3
**Rating:** 4
**Confidence:** 4

**Summary:**

This paper introduces Feedback Forensics, an open-source toolkit designed to measure and analyze the "personality" traits of AI language models—characteristics in model responses such as politeness, verbosity, or tone that influence user experience but are not captured by standard benchmarks. Leveraging both human and AI-generated annotations of pairwise model response data, the toolkit quantifies which traits are encouraged by human (or AI) feedback and which traits are exhibited by various AI models. The authors illustrate their method through a Python API and an interactive web app, releasing not only the toolkit and infrastructure but also underlying annotation datasets. The toolkit aims to provide transparency and actionable insights into model personality, helping researchers and practitioners identify, track, and optimize desirable conversational behaviors across models and feedback-based training regimes.

**Strengths:**

- This paper makes a substantial and timely contribution by introducing an open-source, extensible toolkit for measuring and tracking AI model personality traits.
- Its operationalization of "personality measurement" within the context of AI model development seems signifgant. While prior works often qualitatively discuss model style or provide ad hoc definitions, this toolkit formalizes and systematizes personality trait assessment across datasets and models, drawing on techniques from Inverse Constitutional AI and enriching them with a modular annotation and metric computation pipeline.
- The authors provide both a Python API and a browser-accessible web app, lowering the barrier for practitioners to inspect personality drift or trait encouragement in large models and feedback datasets. The toolkit fills an important methodological gap left by conventional benchmarks, providing the community with a much-needed lens on a critical qualitative dimension of model behavior.

**Weaknesses:**

- The most salient is the lack of direct comparison to existing baselines in the personality measurement domain. There is now a growing body of work proposing frameworks or metrics to quantify AI or LLM "personality", such as evaluations based on the Big Five personality model, or Model Personality Evaluations [1][2]. Without benchmarking Feedback Forensics against such prior methods, it is difficult to clearly assess its incremental value, or to know under what conditions it outperforms, complements, or perhaps fails compared to alternatives.
- The trait taxonomy employed in Feedback Forensics is not extensively justified: it is not clear if the set of personality traits is exhaustive or robust to more subtle variations in conversational style, and there may be gaps in coverage (e.g., cross-cultural or context-dependent personality facets).
- Since the annotators performing trait detection are themselves LLMs, and as these same (or similar) LLMs contribute to both feedback and target models, there is a risk that toolkit outputs merely reflect the biases of the model(s) rather than an independent grounded measurement. This could create feedback loops whereby models overfit to their own interpretations of style and personality. Also, from my point of view, It is unclear how robust trait detection and measurement are to changes in prompt phrasing, dialogue context, or user intent. No ablation studies or error analyses are presented to ensure that detected personality differences are intrinsic to models, and not artifacts of particular prompt distributions.

[1] Evaluating and Inducing Personality in Pre-trained Language Models
[2] https://arxiv.org/abs/2406.17675

**Questions:**

- The current framework treats high-level "personality traits" as discrete, binary, or categorical labels, but many relevant features of AI personality (e.g., confidence, politeness) are multifaceted and can vary on a spectrum. Is there any discussiom about this point?
- How does Feedback Forensics compare quantitatively and qualitatively with prior personality measurement techniques for LLMs (e.g., Big Five surveys, direct style benchmarks, or recent works)? Can you report performance or findings on shared benchmarks?
- How do the personality trait measurements produced by Feedback Forensics translate into actionable improvements in the overall performance or reliability of LLMs, beyond simply characterizing stylistic differences? For example, unlike MBTI-style personality assessments—which primarily offer descriptive labels and do not directly affect task performance or user satisfaction—can your toolkit's results be used to guide model training, calibrate feedback datasets, or optimize model deployment in ways that measurably enhance the utility, safety, or user experience of LLMs? Have you evaluated or do you anticipate any concrete downstream impacts (such as reducing undesirable behaviors or increasing user trust) based on trait monitoring and adjustment?

---

> ### Author Response · Authors · 2025-11-28
>
> Thank you for your detailed review! We address your concerns in our response below.
>
> > **W1:** The most salient is the lack of direct comparison to existing baselines in the personality measurement domain. There is now a growing body of work proposing frameworks or metrics to quantify AI or LLM "personality", such as evaluations based on the Big Five personality model, or Model Personality Evaluations [1,2]. Without benchmarking Feedback Forensics against such prior methods, it is difficult to clearly assess its incremental value, or to know under what conditions it outperforms, complements, or perhaps fails compared to alternatives.
> > [1] Evaluating and Inducing Personality in Pre-trained Language Models
> > [2] https://arxiv.org/abs/2406.17675
>
> **Feedback Forensics intentionally measures different personality traits** to the methods mentioned [1,2] that focus on applying conventional human psychometrics to AI, such as *Big Five* or *Vignette* tests. Our Feedback Forensics toolkit considers a broader definition of personality and focuses on AI-specific personality traits raised in the literature and public discourse around AI models, such as *tone*, *politeness* and *style*. These traits have been shown to affect human annotators and users but are generally not included in human psychometrics tests applied to AI. **Thus, these works cannot be directly compared or benchmarked against Feedback Forensics.** Instead, our toolkit complements this work covering a different set of personality traits.
>
> Beyond human psychometrics, works on *Inverse Constitutional AI* (ICAI) [3] and *VibeCheck* [4] are most similar to our toolkit. These works introduce tools to dynamically discover important traits (including personality traits) in human feedback and models. Feedback Forensics directly builds on these approaches: using part of the ICAI pipeline for its labelling process as well as enabling ICAI-style discovery of new principles, and including a number of personality traits in its default list originally reported in VibeCheck. Due to the dynamic nature of ICAI and VibeCheck, the tested traits vary and are not always directly comparable to Feedback Forensics. Nevertheless, we can observe similar results across these approaches when applied to Chatbot Arena data ([3] Table 2, [4] Fig 2, and Fig 5 in our work). For example, we observe a strong preference for *well-formatted responses* and *confident responses* across all works.
>
> Further note that in our related work discussion (L425-438), we directly qualitatively compare our work to [1] and similar work applying human personality tests to AI models. We had previously missed [2] and have added this to our manuscript. Thank you for highlighting this work to us. Note that we have discussed similar work as shared above.
>
> - [3] Findeis, Arduin, et al. 2025. ‘Inverse Constitutional AI: Compressing Preferences into Principles’. The Thirteenth International Conference on Learning Representations.
> - [4] Dunlap, Lisa,et al. 2025. ‘VibeCheck: Discover and Quantify Qualitative Differences in Large Language Models’. The Thirteenth International Conference on Learning Representations.
>
>
>
> > **W2:** The trait taxonomy employed in Feedback Forensics is not extensively justified: it is not clear if the set of personality traits is exhaustive or robust to more subtle variations in conversational style, and there may be gaps in coverage (e.g., cross-cultural or context-dependent personality facets).
>
> We provide extensive details on the default set of personality traits tested by Feedback Forensics in Section 2.2 and Appendix I.1.1. As we mention there, we do not consider the current list as final (i.e. *exhaustive*) and are planning to keep updating the list as new traits get raised in the public discussion. Yet, no more exhaustive list of AI-model-specific personality traits exists publicly to the best of our knowledge.
>
> Further, our toolkit enables users (1) to bring and evaluate their own personality traits, and (2) automatically discover new relevant personality traits using Inverse Constitutional AI (ICAI). Thus, if a user has a unique context, they are able to adjust the tested personality traits to their needs without relying on our default trait set. Ultimately, Feedback Forensics is a tool intended for configuration by users. We provide some default settings, but users are able to adjust these as needed.

---

> > ### Author Response · Authors · 2025-11-28
> >
> > > **W3 (Part I):** Since the annotators performing trait detection are themselves LLMs, and as these same (or similar) LLMs contribute to both feedback and target models, there is a risk that toolkit outputs merely reflect the biases of the model(s) rather than an independent grounded measurement. This could create feedback loops whereby models overfit to their own interpretations of style and personality.
> >
> > We agree that just relying on LLM annotators contains some inherent risks regarding the quality of the annotations. Thus, we have included a comparison of LLM to human annotations to validate that the results are not strongly affected by LLM biases (Appendix F.2). As part of this rebuttal, we have now extended this human comparison study to two annotators (overall 2k trait-level annotations) and find that, overall, human and AI agreement is high when it comes to our tested personality traits. Indeed, the agreement is higher than human-AI agreement observed in a more general LLM-as-a-Judge setting.
> >
> > > **W3 (Part II):** Also, from my point of view, It is unclear how robust trait detection and measurement are to changes in prompt phrasing, dialogue context, or user intent. No ablation studies or error analyses are presented to ensure that detected personality differences are intrinsic to models, and not artifacts of particular prompt distributions.
> >
> > We would argue that it is difficult to speak of intrinsic model properties in the same sense as with human personality. Nonetheless, models appear to exhibit overall personality tendencies across diverse prompt distributions. We consider analysing a model’s default personality across use-cases and prompts meaningful, as we can observe significant differences across our diverse set of prompts tested (Figure 8).
> >
> > > **Q1:** The current framework treats high-level "personality traits" as discrete, binary, or categorical labels, but many relevant features of AI personality (e.g., confidence, politeness) are multifaceted and can vary on a spectrum. Is there any discussiom about this point?
> >
> > **We disagree with this characterisation of our framework. Our toolkit is able to provide a fine-grained assessment of a model's personality on a spectrum.** Whilst individual annotations over a single response pair are binary in the sense that one model is selected over another, **our aggregation over many such datapoints allows a fine-grained assessment of personality on a spectrum.** We illustrate this fine-grained assessment in our comparison of models in Figures 8 and 18, where we for example observe that *Gemini-2.5-pro* exhibits *confidence* the most (0.54 strength), followed by *Mistral Medium-3.1* (0.31) and GPT-oss-20b (0.22), and that Claude-Sonnet-4 exhibits confidence the least (0.08).
> >
> > > **Q2:** How does Feedback Forensics compare quantitatively and qualitatively with prior personality measurement techniques for LLMs (e.g., Big Five surveys, direct style benchmarks, or recent works)? Can you report performance or findings on shared benchmarks?
> >
> > See our response above to W1. In short, we do not consider these benchmarks to be directly quantitatively comparable but provide additional information about qualitative differences.
> >
> > > **Q3:** How do the personality trait measurements produced by Feedback Forensics translate into actionable improvements in the overall performance or reliability of LLMs, beyond simply characterizing stylistic differences? For example, unlike MBTI-style personality assessments—which primarily offer descriptive labels and do not directly affect task performance or user satisfaction—can your toolkit's results be used to guide model training, calibrate feedback datasets, or optimize model deployment in ways that measurably enhance the utility, safety, or user experience of LLMs? Have you evaluated or do you anticipate any concrete downstream impacts (such as reducing undesirable behaviors or increasing user trust) based on trait monitoring and adjustment?
> >
> > Yes, we anticipate a diverse set of use-cases for Feedback Forensics. In particular, we are excited to see its use to further study the impact of personality on downstream use-cases, an area with limited understanding but notable potential impact. Additional experiments exploring these downstream use-cases are beyond the scope of our paper, but we look forward to future work in these directions. In this paper, we focus on providing a good assessment of personalities, and have thus as part of this rebuttal further improved the validation of our method with extended human annotation experiments and new statistical measures.

---

### Official Review · Reviewer_g7kN · 2025-10-29

**Soundness:** 3
**Presentation:** 3
**Contribution:** 2
**Rating:** 4
**Confidence:** 4

**Summary:**

This paper introduces Feedback Forensics, an open-source Python toolkit, accompanying web application, and annotation dataset designed to measure and track the "personality" of Large Language Models (LLMs). The core methodology involves using AI annotators (i.e., LLM-as-a-Judge) to evaluate pairs of model responses against a set of curated personality traits (e.g., "is more verbose," "makes more confident statements"). The authors demonstrate two primary use cases for this toolkit: analyzing feedback datasets (e.g., Chatbot Arena, MultiPref, PRISM) and analyzing models (e.g., Gemini 2.5 etc).

**Strengths:**

* Solid Implementation and Usability: This is a well-executed project. The authors have delivered not just a concept but a functional, open-source toolkit. Having interacted with the web application, it's clear the tool is polished, intuitive, and provides a valuable, accessible resource for the community to inspect models and datasets directly.

* Comprehensive Experiments: The utility of the toolkit is demonstrated through a comprehensive set of experiments. The analysis spans multiple popular feedback datasets (Chatbot Arena, MultiPref, PRISM) and a wide array of modern, popular LLMs, providing a broad and relevant snapshot of the current landscape.

**Weaknesses:**

* A Measurement Tool, Not an Explanatory Science: The paper is good at describing personality traits but does not offer a deep investigation into why they emerge or how they are encoded, or any other scientific understanding / insights into related problems. The method—collecting pairwise preferences based on predefined, human-articulated traits—feels like a more granular and automated version of existing platforms like Chatbot Arena. It is "forensics" in the sense of identifying a trait, but not in the sense of uncovering the underlying cause (e.g., is a trait an artifact of RLHF, a result of data mixture, or an emergent property of the architecture, or how should we expect models to change under different conditions?).

* Limited Scope on "Control": The work is focused on "measuring" and "tracking." It does not propose new mechanisms for controlling these behaviors beyond the implied (and already standard) method of simply creating a preference dataset for a desired trait and training on it. The contribution is thus more of a high-quality data-gathering and computation tool rather than a new scientific framework for understanding or steering model personality.

**Questions:**

The authors frame many traits in a way that implies a preference. However, many traits are not inherently "bad," just different (like human personalities). Could the toolkit be used to explore the utility of different personality profiles? For example, in a high-stakes domain like finance or medicine, a "cautious" and "less confident" personality that "acknowledges limitations more" might be strongly preferred. How do you see the tool being used to define optimal, domain-specific personalities rather than just tracking a single "better" one?

To follow up, do different personalities imply performance differences in certain capabilities? Do the authors have any evidence or hypotheses about whether optimizing for these "personality" traits has a positive or negative causal effect on a model's underlying capabilities in, for example, complex reasoning, math, or code generation? Are we, as a community, creating "personalities" that we find pleasing but that might mask trade-offs in downstream task performance?

---

> ### Author Response · Authors · 2025-11-28
>
> Thank you for taking the time to review our paper! We appreciate your feedback and address your concerns below.
>
> > **W1:** A Measurement Tool, Not an Explanatory Science: The paper is good at describing personality traits but does not offer a deep investigation into why they emerge or how they are encoded, or any other scientific understanding / insights into related problems. The method—collecting pairwise preferences based on predefined, human-articulated traits—feels like a more granular and automated version of existing platforms like Chatbot Arena. It is "forensics" in the sense of identifying a trait, but not in the sense of uncovering the underlying cause (e.g., is a trait an artifact of RLHF, a result of data mixture, or an emergent property of the architecture, or how should we expect models to change under different conditions?).
>
> > **W2:** Limited Scope on "Control": The work is focused on "measuring" and "tracking." It does not propose new mechanisms for controlling these behaviors beyond the implied (and already standard) method of simply creating a preference dataset for a desired trait and training on it. The contribution is thus more of a high-quality data-gathering and computation tool rather than a new scientific framework for understanding or steering model personality.
>
> We agree with both of these characterizations. Our primary contribution is a general-purpose tool to *describe* model personality, not to explain or control it.
> Though our tool enables such future work (e.g., ablating training stages and observing personality differences or filtering the dataset), we consider this **out of scope** for this paper.
>
> We would like to highlight that the ICLR call for papers calls for **datasets and benchmarks** (our tool has many characteristics of a benchmark and also contributes data) as well as **infrastructure, software libraries, hardware, etc.**. Take the LMSYS-Chat-1M paper [1] as an example, which introduced a new dataset in ICLR 2024. We hope that our contribution of infrastructure, software libraries, datasets and benchmarks (which you have highlighted as a "valuable, accessible resource for the community") is a valuable addition to ICLR.
>
> [1] Zheng, Lianmin, et al. 2024. ‘LMSYS-Chat-1M: A Large-Scale Real-World LLM Conversation Dataset’. The Twelfth International Conference on Learning Representations
>
> > **Q1:** The authors frame many traits in a way that implies a preference. However, many traits are not inherently "bad," just different (like human personalities). Could the toolkit be used to explore the utility of different personality profiles? For example, in a high-stakes domain like finance or medicine, a "cautious" and "less confident" personality that "acknowledges limitations more" might be strongly preferred. How do you see the tool being used to define optimal, domain-specific personalities rather than just tracking a single "better" one?
>
> Yes! We agree that many traits are neither inherently good or bad. Feedback forensics measures personality on many dimensions and leaves interpretation of this measurement to the user, depending on their goal. One such goal is to identify personality profiles preferred for certain domains, as you suggest. This is closely related to our existing **Figure 6**, which shows encouraged and discouraged personality traits across three writing tasks (e.g., professional email communication benefits from a different personality profile than songwriting). We have added an additional investigation for the domains of Health and Machine Learning (Figure 18 in the revised manuscript), showing, for example, that refusal and avoidance is especially dispreferred in Health domains (likely due to models trained to heightened caution in these domains). In both studies effects are generally small, however, showing that many preferred traits are relatively domain-independent.

---

> > ### Author Response · Authors · 2025-11-28
> >
> > > **Q2:** To follow up, do different personalities imply performance differences in certain capabilities? Do the authors have any evidence or hypotheses about whether optimizing for these "personality" traits has a positive or negative causal effect on a model's underlying capabilities in, for example, complex reasoning, math, or code generation? Are we, as a community, creating "personalities" that we find pleasing but that might mask trade-offs in downstream task performance?
> >
> > Yes, we hypothesize that optimizing for certain personality traits can impact the model's underlying capabilities. If the posttraining stages over-emphasize personality, this new objective may come at the cost of capabilities learned in earlier training stages. Overly casual personalities could also plausibly reduce the desired rigor in the reporting of results. These are only hypotheses, however, as testing this would require comparing models that are identical in all *but* their personality, which are not available to the best of our knowledge. Llama-4 Maverick's two releases (Section 3.2.2) likely differ primarily in personality and would be interesting to evaluate, but we lack sufficient details about both checkpoints to further investigate this. We do agree that this would be very interesting future work, however.

---

### Official Review · Reviewer_XfeE · 2025-10-31

**Soundness:** 3
**Presentation:** 3
**Contribution:** 2
**Rating:** 4
**Confidence:** 2

**Summary:**

This paper proposes Feedback Forensics, an open-source toolkit (with a Python API and a Gradio/Web app) for explicitly measuring “AI personality”, stylistic and tone-level traits that are typically only implicitly captured by human-preference evaluations (e.g., Chatbot Arena). The method augments pairwise response datasets with three kinds of annotations: (i) human preference labels, (ii) “target-model” selectors to isolate a specific model’s outputs, and (iii) AI-annotator personality labels that choose which response exhibits a trait more (e.g., verbosity, confidence). It then computes trait-level agreement using relevance, Cohen’s κ, and a combined “strength = κ × relevance” metric to quantify how much a trait is encouraged by feedback or exhibited by a target model. Using this toolkit, the authors (A) analyze trait preferences in Chatbot Arena, MultiPref, and PRISM, and (B) compare trait profiles across popular models and between two versions of Llama-4-Maverick. The paper releases code, an interactive app, and AI-generated annotation data.

**Strengths:**

- **Clear motivation & scope.** The paper tackles an under-explored aspect of alignment (implicit stylistic bias in feedback training) and provides a practical, reproducible framework to study it. The engineering contribution (toolkit + web UI + datasets) is substantial and well-documented.
- **Comprehensive case studies.** Analyses on Chatbot Arena, MultiPref, and PRISM uncover consistent, interpretable patterns, the cross-model comparison and the Llama-4-Maverick public vs Arena contrast demonstrate the toolkit’s utility in post-hoc auditing.

**Weaknesses:**

- **Human validation is minimal.** The entire human–AI agreement study (Appendix F.2) relies on a single author annotator (100 pairs ≈ 1 000 trait-level judgments). While κ ≈ 0.4–0.6 suggests moderate agreement, the absence of multiple raters, confidence intervals, or inter-rater reliability limits claims of external validity.
- **Statistical under-specification.** Metrics are reported as single values without uncertainty estimates or variance analysis. Because strength = κ × relevance is multiplicative and κ depends on class imbalance, small differences may not be significant.

**Questions:**

- Can you expand the human validation beyond the single-author study (e.g., multi-annotators replication on 500 pairs) and report inter-rater κ and human–AI agreement per trait?
- Would the conclusions hold with a different judge model family (e.g., GPT and Qwen models) or when using multi-vote annotation?

---

> ### Author Response · Authors · 2025-11-28
>
> Thank you for taking the time review our paper and provide contructive feedback! Please find our response to your concerns, including new experimental results, below.
>
> > **W1: Human validation is minimal.** The entire human–AI agreement study (Appendix F.2) relies on a single author annotator (100 pairs ≈ 1 000 trait-level judgments). While κ ≈ 0.4–0.6 suggests moderate agreement, the absence of multiple raters, confidence intervals, or inter-rater reliability limits claims of external validity.
>
> We have **expanded our existing user study to two raters** to test inter-rater reliability. As the labeling task is costly and time-intensive, and we have access to limited resources, we were limited in the number of raters and samples we could add. However, we believe that the extended study (Appendix F.2 in the revised manuscript, changes in blue) significantly strengthens our claims. We observe that (1) the added rater has generally even *higher agreement with AI annotators* (see Table 4), (2) the *inter-annotator agreement is similar to the annotator-AI agreement* (Tables 1 and 3), and (3) *agreement with AI annotators remains generally high* (Table 1). Relevance appears more inherently subjective, but choice agreement (whether annotators choose the same item given shared relevance) remains high and is the most important metric for our claims.
>
> > **W2: Statistical under-specification.** Metrics are reported as single values without uncertainty estimates or variance analysis. Because strength = κ × relevance is multiplicative and κ depends on class imbalance, small differences may not be significant.
>
> Thank you for this suggestion, we agree that additional statistical information around our metrics could be very helpful to our users! Thus, **we have now added a new functionality to compute _confidence intervals_ (CIs) and _significance tests_** for the main strength metric. Using this new functionality, we provide confidence intervals and report significance for all strength metric results in the paper. We also make these statistics available via our interface and Python API.
>
> **Method.** We compute the 95% confidence intervals for each strength value using *bootstrapping*, based on 10k samples drawn with replacement from the originally observed pairwise votes. Further, to test for significance, we apply a *one-tailed binomial test*. Given a trait with positive strength, our test considers the null hypothesis that the true underlying probability of the two annotators agreeing is nevertheless at or below chance agreement ($\text{prob}(\text{agree})<0.5$). We reject the null hypothesis at p-values below 5% and correct for multiple simultaneous tests (with *Bonferroni* method). Given a trait with negative strength, we consider the inverse test with a null hypothesis of chance or above chance agreement between annotators.
>
> **Results.** We observe that **most traits reported in the original results shown in the main paper body have significant strength** based on our new tests. The set of significant traits includes all traits in our main *Chatbot Arena* (Fig 5), *MultiPref* (Fig 7), and *Llama-4-Maverick* (Fig 9) experiments shown in the main paper body. In experiments based on smaller dataset sizes we observe more *insignificant* traits. For our experiment *evaluating personality traits across models* (Fig 8), there are some insignificant traits for some models (especially for *Claude-Sonnet-4*). For our experiment with the smallest sample size, the experiment with writing category subsets of Chatbot Arena, we observe many insignificant traits in the original plots. In this experiment sample sizes vary between 213 and 239 votes per category, notably smaller than the 10k samples in many other experiments. We adjust the corresponding plot to include only traits with significant strength on some categories. We also flag insignificant results by graying them out.
>
> Overall, we really appreciated this suggestion and believe that these new additions of confidence intervals and significance tests made our tool meaningfully more useful. In particular, we are excited that these new statistics can help flag to users when insufficient data is available to understand model personality.
>
> > **Q1:** Can you expand the human validation beyond the single-author study (e.g., multi-annotators replication on 500 pairs) and report inter-rater κ and human–AI agreement per trait?
>
> See our response to W1.

---

> > ### Author Response · Authors · 2025-11-28
> >
> > > **Q2:** Would the conclusions hold with a different judge model family (e.g., GPT and Qwen models) or when using multi-vote annotation?
> >
> > The short answer is **yes, the ablation experiments in our paper indicate that the conclusions would mostly hold across these variations**. We include cross-model family and single- vs multi-vote experiments in Appendix F.2. Firstly, the results in Table 1 show that different judge model families overall show similar agreement with humans, though some, including our selected Gemini-2.5-Flash judge have higher agreement with human votes. We selected the model with the best agreement whilst balancing cost. Secondly, we observe in Table 2 that for the selected judge model, the multi- and single-vote experiment have similar performance. This makes single-vote the economically best choice (as multivote requires 3x tokens). Both of these results have now been made more robust thanks to our scaled-up human study. We include the updated tables in our manuscript as well as above as part of our discussion of the new study.

---

### Author Response · Authors · 2025-11-28
**Rebuttal overview**

We would like to thank all reviewers for their detailed and constructive feedback!

We were excited by the reviewers’ recognition of our toolkit’s contribution to the community: reviewer *KruL* described our toolkit software contribution as *"substantial and timely"*, reviewer *g7kN* stated *"the tool is polished, intuitive, and provides a valuable, accessible resource"*, and reviewer *XfeE* stated that the *"engineering contribution (toolkit + web UI + datasets) is substantial and well-documented"*. Thank you for the kind words!

In the following, we briefly summarize the main changes we made as part of this rebuttal in response to the reviewers’ feedback:

1. **Addition of confidence intervals and significance tests.** Following reviewer *XfeE*’s feedback, we have added bootstrapped confidence intervals as well significance tests for all strength values reported in the manuscript. We find that *most results reported in the main experiments of the original manuscript were statistically significant* (with narrow confidence intervals) and flag any results that were not significant in the revised manuscript.

2. **Extension of human annotation study.** Following suggestions by reviewer *XfeE* and *KruL*, we have improved the validation of our AI annotators by adding a new set of human cross-annotations for all datapoints. This new data allows us to more robustly evaluate potential biases of AI annotators by comparing both inter-human and human-AI agreement on the personality annotation task. We find that (1) the added rater has generally even *higher agreement with AI annotators*, (2) the *human inter-annotator agreement is similar to the human-AI annotator agreement*, and (3) *agreement with AI annotators remains generally high*.

3. **Extension of analysis to new domains.** Following the suggestion by reviewer *g7kN*, we have added new results focusing on use of LLMs in the high-stakes *medical* domain as well as in *machine learning*. We find that human annotators in Chatbot Arena particularly dislike *hesitant* and *avoidant* personality traits when judging responses in the medical domain. Such traits may be more prominent in this context due to safety training.

Further, we have updated the manuscript with various clarifications in response to the concerns by reviewers. We have uploaded an updated version of the manuscript with changes highlighted in *blue*.

Overall, we are confident that the changes and discussions added as part of this rebuttal have addressed the reviewers’ main concerns and overall have significantly improved the paper and our toolkit.

---

**Note to AC following changes to review process due to OpenReview leak.** Unfortunately, due to the OpenReview leak in this year’s ICLR process and subsequent policy changes, we had limited opportunity to engage with the reviewers. We hope that the AC and SAC are able to take the notable improvements presented in our rebuttal into account, despite the reviewers being unable to update scores or comment further. We would like to highlight that the reviewers' original rating does not seem to be well aligned with their general praise of our paper and the positive tone of the review texts, all the more after the new improvements based on their feedback.

---

### Meta-Review · Area_Chair_xfRb · 2025-12-25

**Summary:**

The paper introduces Feedback Forensics, an open-source toolkit and web app that explicitly tracks and analyzes AI model personality traits shaped by human or AI feedback, addressing the limitations of existing feedback-based evaluation methods.

The tools are generally well described, well motivated, and supported by comprehensive evaluations; however, several major issues substantially limit the contribution of this work. First, as noted by Reviewer XfeE, human validation is extremely limited—initially relying on a single author annotator and later extended to only two raters during rebuttal—raising serious concerns about the reliability of the evaluations. Second, Reviewer g7kN questions the scientific significance of the work, particularly the lack of insight into underlying mechanisms and how the proposed analyses could guide or control future model design; the authors’ rebuttal frames the work as purely a dataset and benchmark contribution, but this does not convincingly demonstrate how the tools inform next-generation model development. The AC sides with the reviewers, emphasizing that ICLR prioritizes scientific contributions rather than tool demonstrations, and finds this limitation significant. Third, Reviewer KruL highlights that the trait classification framework lacks grounding in established psychological theory (e.g., Big Five personality models), studies a narrow set of traits without cultural considerations, and relies on LLMs as evaluators, introducing confounding biases and rendering the analysis theoretically ungrounded. Reviewer KruL echoes Reviewer g7kN’s concern that the work does not translate evaluation outcomes into actionable insights for improving models. Although the authors respond to these critiques, the rebuttals are largely unconvincing—for example, acknowledging the use of only two human raters further underscores the limited scope—leaving the work lacking in scientific depth and impact.

Overall, given the current status of the work, the AC recommends rejection at this stage, as substantial additional work is still required for it to meet the acceptance threshold.

**Reviewer Concerns:**

The authors partially addressed reviewer questions—e.g., by adding confidence intervals, expanding human annotations, and extending to new domains—but the major concerns highlighted by reviewers remain largely unaddressed, as detailed in the meta-review above.

**Reviewer Scores:**

As reflected in the meta-review above, all three reviewers are unlikely to increase their scores, as the arguments provided by the authors in the rebuttal are not convincing.

---

### Decision · Program_Chairs · 2026-01-26

Reject